# The Use of Particulate Systems for Tuberculosis Prophylaxis and Treatment: Opportunities and Challenges

**DOI:** 10.3390/microorganisms11081988

**Published:** 2023-08-02

**Authors:** Alejandra Barrera-Rosales, Romina Rodríguez-Sanoja, Rogelio Hernández-Pando, Silvia Moreno-Mendieta

**Affiliations:** 1Posgrado en Ciencias Bioquímicas, Universidad Nacional Autónoma de México (UNAM), A.P. 70228, Ciudad Universitaria, Ciudad de México 04510, México; brae@ciencias.unam.mx; 2Instituto de Investigaciones Biomédicas, Universidad Nacional Autónoma de México (UNAM), A.P. 70228, Ciudad Universitaria, Ciudad de México 04510, México; romina@iibiomedicas.unam.mx (R.R.-S.);; 3Sección de Patología Experimental, Instituto Nacional de Ciencias Médicas y Nutrición Salvador Zubirán, Vasco de Quiroga 15, Delegación Tlalpan, Ciudad de México 14080, México; 4CONAHCyT, Instituto de Investigaciones Biomédicas, Universidad Nacional Autónoma de México (UNAM), A.P. 70228, Ciudad Universitaria, Ciudad de México 04510, México

**Keywords:** polymeric nano- and microparticles, delivery systems, tuberculosis prophylaxis, tuberculosis treatment

## Abstract

The use of particles to develop vaccines and treatments for a wide variety of diseases has increased, and their success has been demonstrated in preclinical investigations. Accurately targeting cells and minimizing doses and adverse side effects, while inducing an adequate biological response, are important advantages that particulate systems offer. The most used particulate systems are liposomes and their derivatives, immunostimulatory complexes, virus-like particles, and organic or inorganic nano- and microparticles. Most of these systems have been proven using therapeutic or prophylactic approaches to control tuberculosis, one of the most important infectious diseases worldwide. This article reviews the progress and current state of the use of particles for the administration of TB vaccines and treatments in vitro and in vivo, with a special emphasis on polymeric particles. In addition, we discuss the challenges and benefits of using these particulate systems to provide researchers with an overview of the most promising strategies in current preclinical trials, offering a perspective on their progress to clinical trials.

## 1. Introduction

Tuberculosis (TB), caused by the intracellular bacillus *Mycobacterium tuberculosis* (Mtb), remains one of the most prevalent infectious diseases, representing the leading cause of death from a single infectious agent until the COVID-19 pandemic. Despite the large global drop in the diagnosis and reporting of cases in the pandemic period, it is estimated that 10.6 million people contracted the disease in 2021, of whom 450,000 were rifampicin-resistant cases, and 1.4 million of people died. Compared with the 10 million cases of TB and the 1.2 million deaths reported in 2019, the worrying increase in cases and the setbacks in the quest to end tuberculosis after COVID-19 are evident [1].

Although this pathogen preferentially generates a pulmonary disease, the infection can be disseminated by generating extrapulmonary TB, such as lymphatic, miliary, and central nervous system TB, which represent approximately 15% of all TB infections and are prevalent in immunocompromised patients [2].

The probability of developing TB disease is much higher among individuals with medical conditions that weaken the immune system, such as HIV/AIDS, diabetes, cancer, renal disease, and severe fungal infections; in individuals who have received organ transplantation or tumor necrosis factor alpha (TNF-α) antagonist therapy; or in individuals that have been exposed to alcohol and tobacco abuse, malnutrition, or air pollution [3,4,5]. Recently, new hypotheses derived from meta-analyses have stated that coinfection with SARS-CoV-2, or the use of drugs to treat it, could accelerate the progression of a preexisting TB infection to pulmonary disease, suggesting that coinfection is a predictor of poor prognosis [6,7].

Adding to the complexity, latent TB infection is recognized as the main source of new TB cases, favoring the prevalence of the disease and its high morbidity and mortality in all countries with a high TB burden. This is not a minor problem if we consider that, according to the latest reports, it is estimated that 23% of the world’s population has a latent infection and the diagnostic methods currently available do not allow us to distinguish between a latent infection and active disease [8].

No less important are the factors that have contributed to the increasing emergence of strains that are resistant to the available antibiotics. Some of these factors include the incomplete and variable protection provided by the existing Bacillus Calmette–Guérin (BCG) vaccines against pulmonary TB [9], late diagnosis, the lack of the timely and proper administration of effective drugs, and extensive treatment regimens that have led to poor patient adherence [5]. Consequently, it is extremely important to develop alternatives that increase or reinforce the protective efficacy of the BCG vaccine [10] and therapeutic alternatives to shorten the treatment timespan and ideally decrease the side effects generated by antibiotics [11].

For TB prevention, the strategies have included the design, development, and evaluation of recombinant BCG strains [12,13,14]; the live-attenuated Mtb strain (MTBVAC) [15,16]; other *Mycobacterium* strains, such as *M. vaccae* [17]; subunit recombinant vaccines [18,19,20,21]; and vectorized recombinant vaccines [22,23]. The most recent and promising candidates that have shown evidence of efficacy in animal studies and human trials are summarized in the current TB vaccine pipeline updated as of October 2022 by the TuBerculosis Vaccine Initiative (TBVI) [24]. Treatment strategies have included the use of therapeutic vaccines produced from different *Mycobacterium* strains, such as *M. indicus pranii* and *M. vaccae* [25,26], BCG recombinant strains [12], multiantigenic and multiphasic vectors [27], adjuvanted antigens alone or in combination with nonsteroidal anti-inflammatory drugs (NSAIDs) [28,29], and detoxified cellular fragments of Mtb, such as RUTI [30,31]. Other promising candidates include the use of peptides from different natural sources, such as antimicrobial peptides [32,33] and scorpion venom peptides [34]. Moreover, the use of cytokine gene therapy has also been proven its efficacy and prevented reactivation in experimental TB models [35,36]. No less important are the opportunities that emerge with natural compounds such as flavonoids and lignan aglycones, diferuloylmethane, polyphenols, and aldehydes, among others, which are isolated from plants, fungi, marine species, and bacteria and have shown interesting results alone or in combination with already approved medications [37,38,39].

Among other interesting strategies, the use of particles is becoming more frequent for both TB prophylaxis and therapy [40]. Particles not only protect the molecules that are administered from degradation but also facilitate their controlled and directed release to the target cells, allowing dose optimization [41,42]. They also contribute to overcoming the lack of immunogenicity of subunit vaccines, because many of these particles are inherently immunogenic or can be manipulated to promote enhanced antigenic uptake and processing, mediating adaptive immune responses [43]. As a result of all their properties and benefits, they have been used to formulate vaccines and treatments for TB. Our main objective is to provide a comprehensive perspective on the current state of the preclinical investigation of TB vaccines and treatments formulated with polymeric particles, the challenges and opportunities in the field, and the impact that they could have in future clinical trials.

## 2. Particulate Systems for the Administration of Vaccines and Therapeutics

Particulate systems are important biotechnological tools that have had an enormous impact on biomedical applications, including basic research, imaging, theranostics, and especially therapeutic or vaccine design and delivery [44,45]. They have sizes ranging from nanometers to micrometers and can be manufactured from inorganic materials (i.e., gold, metal oxides, or silica), synthetic or natural polymers (i.e., aliphatic polyesters or chitosan respectively), or synthetic or natural lipids, among other materials [45,46]. Similar to the materials with which they are manufactured, the methods for their loading and functionalization are varied. Some of these methods include adsorption or immobilization onto the surface, dispersion inside the matrix, linking between the matrix and the bioactive molecule, and encapsulation [47].

The materials and preparation methods of particles define their physicochemical characteristics, such as their size, shape, and charge, which in turn define their biodistribution, targeting, release profiles, toxicity, accumulation time, and clearance [48]. Other properties, such as bioavailability, biodegradability, biocompatibility, and bioadhesiveness, are influenced by the intrinsic properties of the particles and their route of administration [49,50]. Consequently, the main challenge is to reach the best combination of materials to obtain the best particles whose properties guarantee their function and safety in vivo.

Currently, almost all routes of administration can be used to deliver particulate systems, including oral, transdermal, intravenous, subcutaneous, topical, intranasal, and pulmonary routes, the last of which is particularly important for TB treatment and prophylaxis because inhalable formulations are the most effective to induce a memory immune response in the lungs [23,51,52].

In addition, particulate systems can be useful to expand the type of immune response generated, considering that the few currently approved adjuvants are effective in inducing antibody responses but are less successful in inducing cell-mediated immunity, which is very important to eliminate intracellular pathogens such as Mtb [53]. In Figure 1, we consider the advantages and the most common characteristics of particles intended for the nasal and pulmonary administration of vaccines and treatments. In subsequent sections, we address the main factors that justify the use of particles to develop TB vaccines or treatments and summarize the most recent preclinical studies with polymeric formulations.

### 2.1. Particulate Systems for TB Vaccine Development

Greater comprehension of the roles that immune cells play in response to Mtb infection is of vital importance for the development of vaccines against this pathogen [54]. For many years, exhaustive efforts have been made to modify, improve, or find an alternative to the BCG vaccine [13,55,56]. This vaccine, the only anti-TB vaccine approved in humans, confers effective protection against disseminated and meningeal TB only in children, with variable protection in adults. The factors that mainly affect its protective efficacy include coinfections with viruses or parasites, comorbidities, environmental factors, intrinsic genetic factors of both mycobacteria and humans, and, importantly, the route of vaccination [57,58,59]. After intradermal vaccination, the BCG vaccine interacts with resident epidermal macrophages, whereas Mtb interacts, in most cases, with resident alveolar macrophages (AMs) and does not suffer opsonization. Consequently, antigenic recognition, uptake, processing, and presentation are different, with implications for the induction of the T-cell memory response required for protection against lung disease [57]. This complex situation has justified the administration of the BCG vaccine directly into the respiratory system as a strategy to induce resident memory T cells in the lung [60,61,62] and the exploration of new vaccines against TB that can be administered by nasal or pulmonary routes, which favor the retention of the antigen at mucosal sites, the induction of systemic and mucosal immunity, and, importantly, the development of lung-resident memory T cells. These are important advantages of mucosal vaccination and, of course, an opportunity for the use of particulate systems [63,64].

#### 2.1.1. Immune Activation Induced by Mtb and Particulate Systems

After inhalation, mycobacteria in the deep lung (alveoli) can interact through pattern recognition receptors (PRRs) with AMs and dendritic cells (DCs). Mycobacterial endocytosis leads to the activation and maturation of these cells and the migration of DCs toward the lung-draining lymph nodes for antigenic presentation and the differentiation of T lymphocytes toward a Th1 type. Th1 cells contribute to the elimination of bacilli and create a positive feedback loop by secreting IFN-γ, which in turn activates more macrophages, enhancing the microbicidal response against Mtb by executing functions including the secretion of microbicidal factors and cytokines such as TNF-α [65,66]. In the same way, particles formulated in prophylactic or therapeutic vaccines can also interact with and activate innate immune cells, increasing their mycobactericidal performance to prevent or combat the infection (Figure 2). Particles can also promote endocytosis by professional phagocytes, induce the production of cytokines and microbicidal factors such as nitric oxide and reactive oxygen species (ROS) [67,68], or induce apoptosis and autophagy [69,70,71], which together are very important mechanisms to eliminate bacilli.

Importantly, these particulate formulations can be administered by several routes, such as parenteral, nasal, and pulmonary, protecting the antigen and supplying it to immune cells, and they can also be engineered to have intrinsic immunostimulant activity that increases the microbicidal performance of cells. In such scenarios, they can act as delivery systems, adjuvants, and immunostimulants, simultaneously or separately, which is highly desirable for the formulation of subunit vaccines against TB. For this purpose, the most used nano- and microparticles include natural and synthetic polymeric capsules and spheres (mainly of chitosan and poly(lactide-co-glycolide) (PLGA)) [72], followed by liposomes and derivatives, solid lipid nanoparticles (SLNs), and immune-stimulating complexes (ISCOMs) [73,74,75]. In Figure 3, we summarize the main functions of particles in vaccines against TB depending on their use as delivery systems, adjuvants, or immunostimulants.

#### 2.1.2. In Vitro and In Vivo Evaluation of Particulate TB Vaccines

When particles are added into a vaccine formulation, in addition to antigens and adjuvants, in vitro preclinical studies are necessary to characterize the particles’ properties, their capacity to transport and release antigens, and their stability, safety, and efficacy in the formulation in terms of the immune response induced in cell lines or primary isolates [76,77]. For instance, one of the most complete in vitro characterization studies was carried out on the subunit vaccine candidate ID93 [76,78]. The authors used the recombinant TB antigen ID93 (composed of three immune-dominant antigens and one latency-associated antigen) conjugated to a modified liposome (mGLA-LSQ). This liposome has intrinsic adjuvant properties because it contains the TLR4 agonist glucopyranosyl lipid adjuvant (GLA) and the saponin QS21. The authors demonstrated that the vaccine was stable and bioactive for 3 months, being able to induce the secretion of IL-2, INF-γ, and TNF-α in a cytokine stimulation assay using fresh whole blood from 10 healthy donors [78]. Most of the time, and if the formulation is successful in vitro, the next step is to test it in vivo. These studies are more robust in exploring the immune response generated after administration by different routes and are a requirement to proceed to clinical phase studies. In the last decade, most of the particulate TB vaccine candidates tested have contained polymeric particles that encapsulate, accompany, or present the antigen on their surfaces and have been administered by the parenteral or mucosal routes. In Table 1, we summarize some recent in vivo studies carried out with particulate TB vaccine candidates based on natural and synthetic polymers, showing the scheme of immunization and the immune response induced.

In contrast to the growing number of preclinical phase studies conducted with particulate TB vaccine formulations, progression to clinical phase trials is scarce. Ongoing clinical trials of new TB vaccines were recently reviewed by Saramago et al. [86]. Based on their review, and in our search, only two particulate vaccine candidates have progressed to clinical studies: ID93+GLA-SE and GamTBvac. Coler et al., conducted a randomized, double-blind phase I study in 60 healthy non-TB-exposed non-vaccinated adults. The purpose was to evaluate two dose levels of the ID93 antigen, administered intramuscularly alone or in combination with two different doses of the GLA-SE adjuvant. The vaccine was safe and well tolerated under all regimes and induced antigen-specific IgG responses in subjects that also received the adjuvant. The use of the adjuvant also enhanced the magnitude and cytokine profile of polyfunctional CD4^+^ T cells [87]. Tkachuk et al., in 2020, conducted a phase II study with 180 healthy volunteers previously vaccinated with BCG and immunized subcutaneously twice at 8-week intervals with their vaccine, GamTBvac. This was a particulate system composed of a multi-antigen fusion protein (the TB antigens Ag85A-ESAT6-CFP10 and a dextran-binding domain) immobilized on dextran NPs and a CpG adjuvant. The vaccine was also safe and well tolerated and induced antigen-specific IFN-γ release, augmented Th1 cytokine-expressing CD4^+^ T cells, and a higher IgG response in vaccinated subjects [88].

### 2.2. Disadvantages of Conventional Treatments for TB and Opportunities for Particulate Formulations

After infection, the main objective is to target the mycobacteria that are present inside macrophages, which the immune system is unable to eliminate. It would also be relevant to target the bacteria that are present inside neutrophils or DCs, with therapeutic agents. However, most of the WHO-recommended drugs for TB treatment, which show variable permeability, are administered by oral or intravenous routes, implying that they are present at high concentrations in serum but not in the lungs. This partially explains their lower effectiveness in pulmonary disease treatment and their higher toxicity [89]. Additionally, Mtb not only survives inside the cells but also in the granuloma, the complex multicellular structure formed as a result of the host immune response, and drugs must also permeate these structures and reach the mycobacteria that are contained within them [90]. Importantly, prolonged treatments for drug-susceptible TB (6 months of isoniazid and rifampicin, with the addition of pyrazinamide and ethambutol in the initial 2 months) and drug-resistant TB (between 9 and 24 months depending on the strain) become very toxic, increase secondary adverse effects, and therefore decrease patient adherence to the treatment scheme [91,92].

One way to overcome or at least partially resolve these problems is to use engineered carriers for the directed administration of TB drugs, such as liposomes, solid lipid nanoparticles (SLNs), and polymeric micro- and nanoparticles [89]. The physicochemical characteristics of these particles, mainly but not exclusively the size, surface charge, and functionalization, are crucial in the design and must be considered together with the route of administration. Particularly for the treatment of TB, the inhalation route is of interest to target resident AMs loaded with bacteria. In this regard, several investigations have been developed to find the ideal characteristics that a particle must have to reach this cell population and deliver its load (Figure 1) [93]. Another important advantage of this route of administration is that it mimics the course of bacterial spread: because the AMs are the first cells to phagocytize Mtb and drug-containing particles upon inhalation, they traffic them to the lung interstitium and travel to the site at which the bacteria tend to migrate, which can guarantee the directed and controlled release of anti-TB drugs and, consequently, a more precise dosage with fewer side effects [94].

#### 2.2.1. In Vitro Evaluation of Particulate TB Drug Delivery Systems

In the same way as for evaluating particulate vaccines, in vitro assays are also required for the preclinical investigation of anti-TB treatments formulated with particles. Each study has its limitations and advantages, but they are essential in determining the safety and efficacy of these systems. In vitro studies are also very important to standardize and achieve particles with an optimal aerodynamic diameter for pulmonary delivery, ensuring deposition in the apical and deep regions of the lung [95]. These studies are also critical in characterizing the physicochemical properties, stability, loading efficiency, and release of anti-TB drugs, as exemplified in the works of Garg et al. and Desai et al. [96,97]. Additionally, they allow us to evaluate the phagocytosis of loaded particles, their intracellular accumulation, and their cytotoxicity, because the main objective is to induce lower cytotoxicity than that induced by free drug administration [98,99,100,101,102].

Novel tools such as the in silico stochastic lung model have also been developed to correlate with in vitro studies and to predict the amount of drug deposited quantitatively in the lungs. Mukhtar et al., who reported the fabrication and characterization of a chitosan/hyaluronic acid nanoparticle and isoniazid suspension, predicted, with this model, that a very low fraction of particles was exhaled, while the particle deposition was high in the lung–bronchial and acinar regions, correlating with their in vitro observations [103].

Interestingly, several authors have also evaluated in vitro drug-free microparticles as a strategy to reduce the bacillary load in infected cell lines. For instance, Lawlor et al. reported the use of PLGA particles to reduce the bacillary load in THP-1-derived macrophages infected with the H37Rv strain. Without altering cell viability and without modifying proinflammatory cytokine secretion, they demonstrated that the particles induced NF-kB activation and autophagy in a dose-dependent manner, which in turn increased the killing performance of macrophages [70]. Bai et al., used curcumin particles to treat human alveolar and THP-1-derived macrophages before infection with the H37Rv strain, and curcumin also reduced the bacillary load through the induction of autophagy and caspase-3-dependent apoptosis [69]. Machelart et al., using beta cyclodextrin NPs, demonstrated that they were efficiently captured by bone-marrow-derived macrophages and bone-marrow-derived dendritic cells and were able to impair Mtb replication and induce apoptosis in infected macrophages [71].

In Table 2, we summarize some studies carried out in the last decade with particulate TB drug delivery systems based on natural and synthetic polymers, tested on different cell lines before or after infection with mycobacteria.

Although less frequent, the study of inorganic particles that have a direct microbicidal effect is also of interest. Gold and silver NPs have been functionalized with variable ligands, such as citrate or polyallylamine hydrochloride A, to effectively reduce the cell viability of mycobacteria [113]. The antimycobacterial properties of gold have been supported by auranofin, a gold-based antirheumatic drug that inhibits bacterial thioredoxin reductase, making replicating and nonreplicating mycobacteria susceptible to oxidative species; consequently, gold has become a suitable material to develop particles for the treatment of TB [114,115].

#### 2.2.2. In Vivo Evaluation of Particulate TB Drug Delivery Systems

In vivo studies conducted to evaluate particulate TB drugs are performed with antibiotic-loaded particles (against drug-sensitive and drug-resistant Mtb strains) and are useful in showing prolonged drug release, long-term antibacterial effects, reduced toxicity, and the prevention of infection relapse. There is agreement that, for inhalable formulations, the most appropriate materials are natural or synthetic polymers, and those made from polysaccharides are especially promising. Wu et al. evaluated the in vivo toxicity and release properties of an inhalable preparation of chitosan nanogel particles loaded with genipin, isoniazid, and rifampicin. They demonstrated enhanced antimycobacterial activity in mice infected with the resistant H37Rv strain [116]. Machelart et al., with their beta-cyclodextrin NPs administered by direct aerosolization, were also able to decrease the Mtb burden in the lung after infection, and the authors proposed that this observation was a result of AMs’ reprogramming by these particles, which had intrinsic immunostimulant properties [71].

Grehna et al., showed that after the pulmonary administration of spray-dried locust bean gum MPs loaded with isoniazid and rifabutin, lung infection and mycobacterial growth rate values were decreased in the spleens and livers of infected mice. The short-term treatment regimen (five times per week) that the authors used was more effective than the oral coadministration of both antibiotics, even at lower doses. Additionally, they highlighted that polysaccharide-based particles are promising for pulmonary administration because they contain sugar units that are recognized by surface receptors expressed by AMs [117]. Singh et al. in 2021 also developed a dry powder for inhalation, composed of 25% isoniazid, 25% rifabutin, and 50% biodegradable polymer poly(L-lactide). The authors demonstrated the efficacy, safety, and tolerability of the inhalable particles in three TB models (high-dose intravenous and low-dose aerosol infection in mice and low-dose aerosol infection in guinea pigs). They were also able to prevent the relapse of infection four weeks after stopping the treatment, using the combination strategy of half the oral dose of antibiotics with inhalable particles [118]. Antonov et al. showed that encapsulated levofloxacin in PLGA MPs achieved greater bacterial clearance than the free drug orally administered after infecting mice with the H37Rv strain. The particles demonstrated suitable biocompatibility and release kinetics [119].

In contrast to the growing number of preclinical phase studies conducted with particulate formulations for TB treatment, progression to clinical phase trials is also scarce and, based on our search, there are no polymeric formulations at this stage of investigation. Srichana et al., demonstrated the safety of a dry powder formulation with liposomes containing four anti-tuberculosis drugs (isoniazid, rifampicin, pyrazinamide, and levofloxacin) administered via inhalation to 40 healthy adults. After successfully passing this clinical phase I trial [120], the formulation was evaluated for approximately eight weeks in 44 adult patients with active pulmonary TB. Although the treatment did not increase Mtb sputum culture conversion after two months, the percentage of patients having adverse side effects was significantly lower. The main results were decreased cough at 4 weeks of treatment, substantially reduced hemoptysis at 2 weeks of treatment, and lower incidences of nausea and vomiting [121].

#### 2.2.3. Opportunities for Particulate Systems for TB Theranostics

Recent studies have focused their attention on theranostics as means to combine early diagnosis and the administration of targeted treatments in a single system. Particulate systems applied to TB theranostics must be developed with a favorable aerodynamic diameter for pulmonary delivery, to maximize drug delivery while avoiding toxic systemic side effects and potentially shortening the treatment duration. These systems are composed of a biocompatible metal organic framework (MOF) as a drug carrier, which usually has synergistic therapeutic activity and one or several anti-TB drugs [122]. The MOF delivers its cargo upon activation by endogenous stimuli such as pH, redox, or ATP or by exogenous stimuli such as temperature, ions, pressure, light, humidity, or a magnetic field [123]. Recently, Jiménez-Rodríguez et al. successfully encapsulated RIF in liposomes and silver nanoparticles to develop a luminescent biomarker for its evaluation as a TB theranostic. The particles permitted early diagnosis and treatment, and, due to their optical properties, the authors highlighted their utility in pharmacokinetic studies [124].

An emerging opportunity for TB theranostics is the tracking of complex structures such as granulomas and encapsulating various anti-TB drugs for directed administration. In latent TB that can become active TB, this strategy is a priority because granulomas contribute to the persistence and/or spread of the bacilli present inside them. For this reason, in recent studies, the use of sophisticated systems to localize and treat early granulomas has been explored. Liao et al. designed a TB granuloma imaging-guided photodynamic therapy (PDT) using an aggregation-induced emission carrier. After exposure to white light, the carrier generated ROS and simultaneously released rifampicin. With this system, the authors were able to perform an early diagnosis ex vivo using a granuloma tail model in mice and control the drug-sensitive and drug-resistant bacteria in vitro [125,126]. However, these strategies are in the preliminary stage of investigation and their efficacy and safety levels need to be further studied and characterized.

In Figure 4, we provide an overview of the main advantages, disadvantages, challenges, and opportunities regarding particulate systems for the formulation of TB vaccines or treatments.

## 3. Concluding Remarks and Prospects

In addition to the challenges involved in combating the immune evasion and resistance mechanisms generated by mycobacteria, challenges related to the development of novel and safe protective and therapeutic alternatives against TB persist. A sophisticated approach that matches the sophisticated evasion mechanisms of Mtb is needed to target infected cells or cells that potentially will be infected. One important approach is the use of particles. They are not only ideal for pulmonary administration, imitating the portal of entry and path of bacilli, but can also be engineered by selecting the most desirable materials, ligands, and physicochemical characteristics depending on whether it is a vaccine or a drug delivery system.

However, the landscape is complicated because few studies have moved on to clinical research phases. This may be because, despite the versatility and advantages of particles, they also have intrinsic limitations, and their performance is affected by external factors, which must be weighed against the benefits, especially regarding to formulating a vaccine or a treatment for a disease of such relevance as TB. Complicating the academic scenario, other associated problems that likely affect translation to clinical practice are related to the lack of clinical phase III trials that evaluate the efficacy and safety of these systems in large populations, the availability of resources and infrastructure for research and development, and the consideration of the health emergency that TB represents globally, especially in countries with a higher prevalence and the presence of hypervirulent multi-drug-resistant strains.

As researchers involved in the field of particulate vaccine development for mucosal administration, we aimed to provide a grounded perspective on the importance of the rational design of these systems, since they are recognized, phagocytosed, and can influence the same cells that the tubercle bacillus interacts with. Moreover, we sought to highlight the value of multidisciplinary work to advance the field and the work that is required to move beyond a proof of concept without losing sight of the challenges imposed by particles and Mtb in its progression.

## Figures and Tables

**Figure 1 microorganisms-11-01988-f001:**
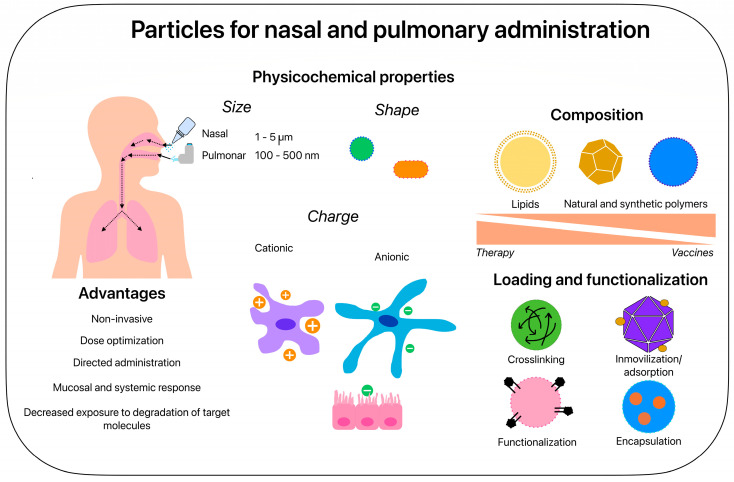
Particles for nasal and pulmonary administration. The main physicochemical properties to consider for the preparation of particles intended for TB prophylaxis and treatment are size, shape, and surface charge. Sizes from 100 to 500 nm are preferable but not exclusive for nasal administration, while 1–5 μm particles are optimal for intrapulmonary administration. Shapes such as those present in nature are also favorable for the internalization of particles in the lung, mainly those that are spherical or rod-shaped, as is the case for Mtb. Additionally, a preferential but not exclusive interaction between cationic microparticles (+) and macrophages and anionic nanoparticles (−) with dendritic and epithelial cells has been documented. All these properties in turn will depend on the fabrication materials. For the formulation of TB vaccines, the most reported materials are natural and synthetic polymers, followed by lipids (for the fabrication of solid lipid nanoparticles and liposomes), while, for TB treatment formulations, the use of lipids is most frequent, followed by synthetic and natural polymers.

**Figure 2 microorganisms-11-01988-f002:**
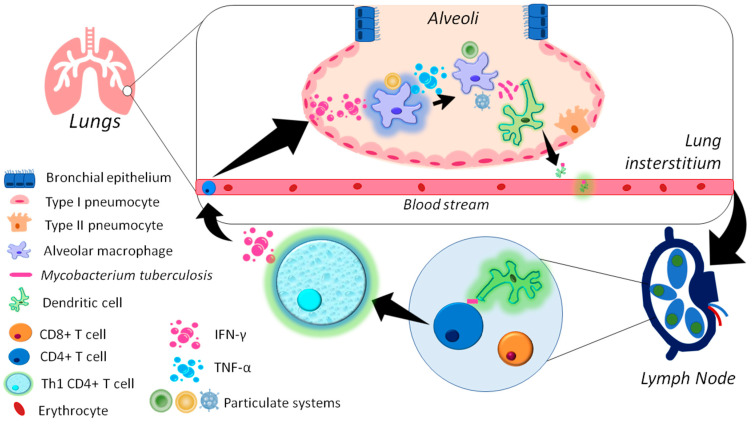
Immune activation induced by Mtb and particulate systems after inhalation.

**Figure 3 microorganisms-11-01988-f003:**
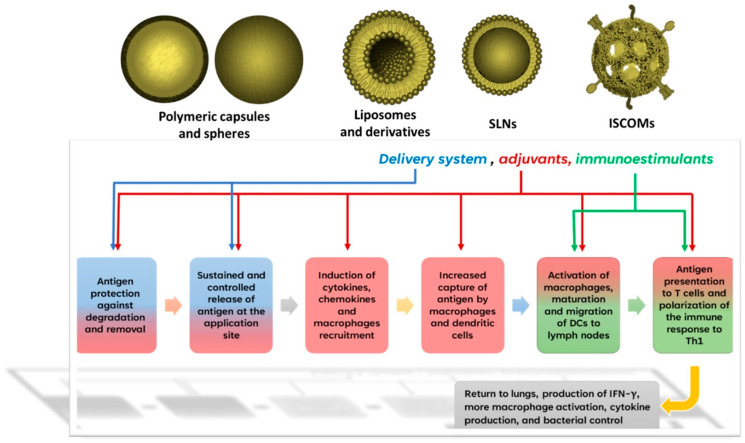
Roles of particles in the formulation of vaccines against TB.

**Figure 4 microorganisms-11-01988-f004:**
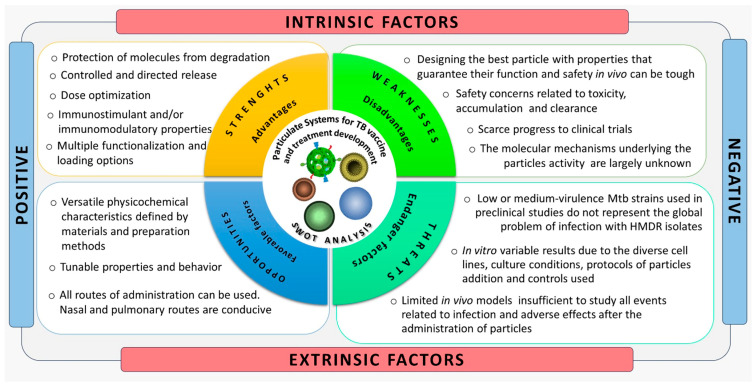
SWOT analysis of particulate systems for the formulation of TB vaccines or treatments. HMDR: hypervirulent multi-drug resistance.

**Table 1 microorganisms-11-01988-t001:** Polymeric particulate TB vaccines and the immune response induced in vivo.

Particulate System		Vaccine Formulation (Antigen/Adjuvant)	Scheme of Immunization (Model/Route/Dose)	Immune Response Induced	Ref
NATURAL POLYMERS	Chitosan NPs *Encapsulation*	pDNA encoding Ag 85B No extra adjuvant	BALB/c mice 50 μg SC, 1× at day 0 IN, 2× at 2w interval	- Higher levels of Ag-specific IgG and IgG2a - Enhanced proliferation of CD4^+^ T cells - Higher release of IFN-γ and IL-2 in splenocytes restimulated with Ag 85B - Induction of autophagy	[79]
	Chitosan NPs *Mix*	ESAT-6 (1–20 peptide) A: MPL	C57BL/6J, IFN-γ^−/−^ and IL-17^−/−^ mice 133 μg/50 μg IN, 3× at 2w interval	- Potent induction of Th17 in lung and Th1/Th17 in spleen - Higher protection against Mtb infection - Induction of NLRP3-independent inflammasome and IL-1β	[80]
	Chitosan NPs *Coating*	Mtb cell wall lipids No extra adjuvant	BALB/c mice 0.5 mg/kg SC, 1× IP, 4× 0, 21, 45, 66 days	- Higher levels of Th1 (IFN-γ, IL-2, TNF-α) and Th2 cytokines (IL-4, IL-5, IL-13) in LNs and spleen - γδ T-cell activation in LNs - Higher levels of IgG, IgG1, IgG2, and IgM	[81]
	Inulin chitosan NPs *Conjugation*	Fusion CT (CFP10-TB10.4) No extra adjuvant	C57BL/6 mice 100 μg/mL SC, 3× 0, 14, 28 days	- Higher release of IFN-γ, TNF-α, IL-2, and IL-4 in splenocytes restimulated with CT - Higher levels of CT-specific IgG1 and IgG2b	[82]
	Advax™ (δ-Inulin-NPs) *Mix*	Fusion CysVac2 (Ag85B-CysD) A: Advax CpG	C57BL/6 mice 3 μg of fusion/mg inulin IM, 3× at 2w interval	- Induction of specific multifunctional CD4^+^ T cells (IFN-γ^+^, TNF^+^, IL-2^+^) - Reduction in CFU in lung after Mtb infection - Strong immunogenicity and protection	[67]
	Advax™ (δ-Inulin-NPs) *Mix*	Fusion CysVac2 (Ag85B-CysD) A: Advax	C57BL/6 mice 3 μg of fusion/mg inulin IT, 3× at 2w interval	- Induction of lung-resident antigen-specific IL17-secreting CD4^+^ cells - Higher protection against Mtb infection compared with BCG-vaccinated mice	[63]
	INU/pArg NCs *Adsorption*	Fusion ECH (ESAT6/CFP-10) A: Imiquimod	C57BL/6 mice 10 μg of fusion IN, 3×	- Higher levels of IgA in bronchoalveolar fluid - Higher titers of IgG in sera - Higher release of IFN-γ and IL-17 in splenocytes restimulated with ECH	[83]
	Dextran NPs *Immobilization*	Fusion GamTBvac (Ag85A-ESAT6-CFP10-DBD) A: DEAE-dextran-CpG	C57BL/6 mice and guinea pigs 5, 10, 20 μg of fusion SC, 2× at 3w interval	As booster of BCG vaccine in mice: - Higher levels of IFN-γ and Ag-specific IgG - Reduction in CFU in lung - Higher survival	[68]
SYNTHETIC POLYMERS	PLGA NPs *Encapsulation*	Fusion HspX/EsxS A: DOTAP	BALB/c mice 25 μg of fusion/5 mg NPs SC, 3× at 2w interval	- Higher levels of IFN-γ - Higher titers of specific IgG1 and IgG2a compared with BCG	[84]
	PLGA NPs *Encapsulation*	Plasmid pcDNA3.1/Mtb72F A: TB10.4 and/or CpG	BALB/c mice SC, 1× BCG or plasmid at day 0 SC, 3× 7, 14, 21 days	As booster of BCG vaccine: - Higher levels of IFN-γ in splenocytes restimulated with BCG	[85]
	Polyester NPs *Coating*	Fusion H28 (Ag85B-TB10.4-Rv2660c) H4 (Ag85B-TB10.4) A: DDA	C57BL/6 mice 2–10 μg of fusion SC, 3× at 9-day intervals	PNPs-H4 induced: - Long-lasting antigen-specific T-cell responses - Protective immunity in infected mice - Reduction in CFU in lung - Similar protective immunity to BCG	[43]

NPs: nanoparticles; INU/pArgNCs: inulin/polyarginine nanocapsules; PLGA: poly(lactide-co-glycolide); A: adjuvant; MPL: monophosphoryl lipid A; DEAE: diethylaminoethyl; DOTAP: 1,2-dioleoyl-3-trimethylammonium propane; DDA: dimethyldioctadecyl ammonium bromide; SC: subcutaneous; IN: intranasal; IM: intramuscular; IP: intraperitoneal; IT: intratracheal; LNs: lymph nodes.

**Table 2 microorganisms-11-01988-t002:** Particulate systems evaluated in vitro as carriers of anti-tuberculosis drugs.

	Particulate System	Drug	Administration Scheme (Cell Line/Strategy)	Observations	Ref
NATURAL POLYMERS	Chitosan MPs	INH RFB	A549, THP-1 Mφ AI with Mb BCG	- Cell viability above 70% for A549 cells - Dose-dependent effect on THP-1 Mφ - Microencapsulation preserved antibacterial activity of drugs - Free and drug-loaded MPs induced increased secretion of TNF-α and IL-18 in THP-1 Mφ	[104]
	Chitosan NPs	Anti-Cystatin C siRNA	HMDM, THP-1 Mφ BI with Mtb H37Rv and susceptible and resistant isolates	- Loaded NPs were non-cytotoxic and were efficiently internalized by cells - Significant reduction in intracellular bacteria	[105]
	Fucoidan MPs	RFB INH	A549, THP-1 Mφ AI with Mb BCG	- Cell viability above 65% at 24 h - Encapsulation reduced RFB cytotoxicity - Free and loaded MPs induced TNF-α and IL-8 - Dose-dependent uptake of MPs	[106]
	Glucan NPs	RFB	J774 AI with Mtb H37Ra	- Induction of ROS and NO within infected Mφ - Induction of lysosome accumulation and phagolysosomal maturation in infected cells - The efficacy of RFB was enhanced 2.5-fold	[107]
SYNTHETIC POLYMERS	PLGA NPs	RIF	RAW 264.7, BMDM BI with Mb BCG	- Loaded NPs promoted the efficient clearing of BCG infection over a 12-day period	[108]
	PLGA NPs	RIF INHP	HMDM BI with Ms	- Sustained release of drugs over 15 days - Six-fold increase in therapeutic efficacy - Higher cell uptake and better antimicrobial activity than free drugs	[109]
	PLGA NPs encapsulated inside MAAEA MPs	RIF	Caco2, MH-S AI with Mtb H37Rv	- Loaded NPs translocated to the basolateral side of Caco2 cells and were not cytotoxic - Loaded and empty NPs decreased growth of intracellular bacteria	[110]
	Poly(ε-caprolactone)	INH SQ641 + CsA + VE	J774A.1 AI with Mtb H37Rv	- Better inhibition of intracellular replication of Mtb with SQ641-CsA-VE than SQ641 alone or INH	[111]
	Poly(ethylene sebacate) NPs	RIF-CUR	RAW 264.7 AI with Mtb H37Rv	- NPs were non-cytotoxic - Showed 1.5-fold higher drug internalization compared to free drugs - Significant killing of intracellular bacteria	[112]

NPs: nanoparticles; MPs: microparticles; PLGA: poly(lactic-co-glycolic) acid; MAAEA: methacrylic acid–ethyl acrylate copolymer; RIF: rifampicin; INHP: pentenyl–isoniazid; INH: isoniazid; SQ641 + CsA + VE: natural analogue of capuramycin + cyclosporine A + vitamin E; RFB: rifabutin; CUR: curcumin; BI: before infection; AI: after infection; BMDM: bone-marrow-derived monocytes; HMDM: human-monocyte-derived macrophages; Ms: *Mycobacterium smegmatis*; Mb: *Mycobacterium bovis*.

## Data Availability

Not applicable.

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
