# Peer review of "The Use of Particulate Systems for Tuberculosis Prophylaxis and Treatment: Opportunities and Challenges"

_microorganisms, 2023, doi:10.3390/microorganisms11081988_

Round 1

Reviewer 1 Report

The figures are of low quality, particularly figure 1 is unprofessional, it is recommended that the figures be made by a publishing professional

Reviewer 2 Report

It has been a pleasure to read this interesting article that reviews the state of the art regarding new prophylaxis and treatment strategies in tuberculosis based on particulate systems. It has allowed me to learn about concepts that are distant from a researcher with a more clinical background. However, some aspects are challenging to follow for someone without experience in this area, and I believe that the manuscript would benefit from a revision to make it more accessible to clinicians. I would like to make some suggestions that I think can contribute to improving the understanding of the manuscript.

Introduction:

In the first paragraph, I would focus on the epidemiology and challenges related to TB without including anything about particulate systems. It could include a mention of the concerning increase in cases after COVID-19. Additionally, it is important to add that there are increasingly more immunosuppressed individuals at risk of TB, such as transplant recipients and those undergoing biological treatments, a challenge for diagnosis and management.

It is essential to mention the difficulties in differentiating active disease from latent infection before discussing the emergence of resistances, which is also important. This should be considered a priority, with the development of tests capable of discriminating or predicting the risk of progression.

Paragraphs 67-78 overlap with the previous one and become repetitive; they could be merged into a single paragraph that reviews the different strategies under development, organized based on their objectives (prophylaxis, treatment, etc.).

Starting from line 79, it would be convenient to begin by defining what particulate systems are, but I would include it in the next section, transitioning from the general concept to their specific use in relation to TB. The last paragraph could be omitted as, by this point, we are already immersed in discussing the epidemiology and challenges.

Line 132 could perhaps be titled "Physiopathology of Recognition or Immunopathogenesis of TB," explaining the generalities first and then moving on to how the recognition of particulate systems works, without involving anything related to vaccines. The following paragraph could then focus on alternative approaches to enable mucosal immunization.

Regarding line 198, I think it needs a grammar review.

Line 194, before the table, it would be helpful to comment in general terms on how these studies are grouped, their objectives, and main results, providing an overview of the current situation. The table is filled with acronyms that may be difficult for clinicians to follow; an explanation of the different compounds is needed beforehand.

The paragraph starting at line 161 does not connect well with the previous one. It is unclear why we suddenly discuss the vaccine. Perhaps this general paragraph about the limitations of BCG should go in the introduction, and here, we should explain how these new particulate systems could be helpful.

Section 2.2:

Some ideas from the introduction could be included in the first paragraph to explain to the reader the interest of particulate systems in treatment. Table 2 seems unnecessary.

The paragraph starting at line 250 could benefit from including some general conclusions about the most promising treatments among those included in table 3, which is complex and, again, filled with acronyms, just like the summary of drug-free microparticles at the end of the paragraph. A summary table of in vivo studies could be included instead of table 2, and the discussion of the most promising findings could be reduced in the text.

2.2.1: The concept/definition of "theranostics" should be mentioned ahead, along with some ideas of what will be discussed later.

Section 3: Since there isn't much information about clinical trials, they could be included in the previous sections, following the order of in vitro, in vivo, and clinical trials (if any) to simplify.

Conclusions:

For the first time, the limitations of these new systems are mentioned here, but they have not been discussed before. It would be good to include them in some section, a specific section maybe, or perhaps in the first section that defines and summarizes their potential uses. The limitations related to animal models mentioned here have barely been developed in the corresponding sections. To conclude, a sentence regarding the author’s guess on the more immediate translation to clinical practice would be great.

I am not a native speaker, but the English sounds good. Some sentences are too long.
